# Resveratrol against Cardiac Fibrosis: Research Progress in Experimental Animal Models

**DOI:** 10.3390/molecules26226860

**Published:** 2021-11-13

**Authors:** Dongmin Yu, Zhixian Tang, Ben Li, Junjian Yu, Wentong Li, Ziyou Liu, Chengnan Tian

**Affiliations:** 1Department of Breast Surgery, First Affiliated Hospital of Gannan Medical University, Ganzhou 341000, China; 18370955568@163.com; 2Department of Cardiovascular Surgery, The First Affiliated Hospital with Nanjing Medical University, Nanjing 210029, China; liben1966@163.com; 3Department of Cardiothoracic Surgery, First Affiliated Hospital of Gannan Medical University, Ganzhou 341000, China; tzhixian2020@gmu.edu.cn (Z.T.); yjj19860228@163.com (J.Y.); tongge87821@sina.com (W.L.)

**Keywords:** resveratrol, cardiac fibrosis, cardiac fibrosis-related diseases

## Abstract

Cardiac fibrosis is a heterogeneous disease, which is characterized by abundant proliferation of interstitial collagen, disordered arrangement, collagen network reconstruction, increased cardiac stiffness, and decreased systolic and diastolic functions, consequently developing into cardiac insufficiency. With several factors participating in and regulating the occurrence and development of cardiac fibrosis, a complex molecular mechanism underlies the disease. Moreover, cardiac fibrosis is closely related to hypertension, myocardial infarction, viral myocarditis, atherosclerosis, and diabetes, which can lead to serious complications such as heart failure, arrhythmia, and sudden cardiac death, thus seriously threatening human life and health. Resveratrol, with the chemical name 3,5,4′-trihydroxy-trans-stilbene, is a polyphenol abundantly present in grapes and red wine. It is known to prevent the occurrence and development of cardiovascular diseases. In addition, it may resist cardiac fibrosis through a variety of growth factors, cytokines, and several cell signaling pathways, thus exerting a protective effect on the heart.

## 1. Introduction

With continuous improvement in the quality of life, cardiovascular and cerebrovascular diseases have emerged as the primary killers. Myocardial infarction and coronary heart disease are the major causes of death associated with several heart-related organic and functional diseases. Therefore, effective preventive strategies and treatment of cardiovascular and cerebrovascular diseases are necessary for a healthy life.

Cardiac fibrosis is a pathological process involved in the remodeling of extracellular matrix (ECM) and excessive accumulation of collagen related to myocardial cells [1]. A large number of studies have confirmed that myocardial fibrosis is a prognostic factor for adverse cardiac outcomes in different cardiovascular diseases [2,3,4,5,6]. Myocardial fibrosis is most common in ischemic injury, but it may also occur in non-ischemic causes, including pressure or volume overload, diabetes, hypertrophic cardiomyopathy (HCM), idiopathic dilated cardiomyopathy (DCM), sarcoidosis and myocarditis [7]. Complex cellular and molecular mechanisms are believed to underlie the development of cardiac fibrosis, as evident from the participation of numerous factors in its occurrence and development, including the renin–angiotensin–aldosterone system (RAAS), endothelin (ET), nitric oxide (NO), growth factors (GFs), calcium ion (Ca^2+^), and tissue inhibitors of metalloproteinase (TIMPs) [8,9].

Resveratrol, whose chemical name is 3,5,4′-trihydroxy-trans-stilbene, also known as stilbene, is a non-flavonoid polyphenol that is widely found in grapes, giant knotweed, peanuts, and other plant species. It exhibits protective functions against cardiovascular and cerebrovascular diseases, thus protecting the nerves, delaying aging, resisting oxidation, inhibiting tumor formation, and reducing inflammation, diabetes, obesity, and fibrosis. Thus, it is a highly researched compound. In addition, resveratrol-related products have been used in treating several diseases [10]. In recent years, with food and drug safety awareness, resveratrol has attracted considerable attention due to its edible and medicinal values. Although the anti-cardiac fibrosis effect of resveratrol has gradually attracted increasing attention, its mechanism is still unclear. In this study, we reviewed the proposed mechanism of resveratrol against cardiac fibrosis (Figure 1) to provide a theoretical reference for further studies on its functions in cardiac fibrosis-related diseases.

## 2. Cardiofibrosis and Its Pathogenesis

Cardiac fibrosis is characterized by an excessive accumulation of ECM and collagen in the myocardial interstitium, a process also known as ECM remodeling. ECM remodeling is an adaptive response of the myocardium to several pathological stimuli (such as hypertension and myocardial infarction), and it is also a common pathological change when many cardiovascular diseases (such as hypertension, myocardial infarction, heart failure, and arrhythmia) develop to a certain stage. Pathologically, myocardial fibrosis is mainly characterized by increased collagen in myocardial interstitium, unbalanced proportion, and disordered arrangement of collagen components, while, functionally, it is mainly characterized by increased myocardial stiffness, ventricular systolic and diastolic dysfunction, and abnormal coronary reserve function. At present, its pathogenesis is not completely clear; it may involve many aspects such as immune regulation, oxidative stress, environmental toxin, and gene mutation [11].

Endothelial to mesenchymal transition (EndMT) is emerging as an important contributor in the development of fibrotic cardiovascular diseases as well as diabetes-associated cardiac fibrosis [12,13]. Moreover, the occurrence and development of myocardial fibrosis is closely related to immune inflammatory cytokines. For example, when myocardial apoptosis and necrosis are replaced by fibrous scars, or when collagen fibers appear in the normal myocardial space, intracellular chemokines are released, resulting in an oxidative imbalance and increased levels of inflammatory cytokines, such as tumor necrosis factor α (TNF-α), interleukin 1 (IL-1), and interleukin 6 (IL-6), which promote fibroblasts to form and secrete ECM macromolecules, and accelerate the progress of cardiac fibrosis [14]. Moreover, it has also been found that transforming growth factor β (TGF-β) can promote myocardial fibrosis and induce the expression of ECM proteins, such as collagen and fibronectin [15,16]. Moreover, a local injection of TGF-β may even promote the formation of fibrous scar granulation tissue [17]. Another study found that, when an acute coronary occlusion occurs, the myocardium in the blood supply area undergoes focal necrosis, followed by conversion of fibrosis into scar tissue. Interestingly, fibrosis is accompanied by a high expression of Stromal Cell Derived Factor-1(SDF-1), suggesting that SDF-1 is closely related to cardiac fibrosis too [18].

## 3. Anti-Cardiac Fibrosis Effect of Resveratrol

Resveratrol is known to prevent the occurrence and development of cardiovascular diseases. In vivo, Zou et al. [19], after treating male wild-type C57BL/6 mice with resveratrol (5 or 50 mg/kg/d, Selleck, Houston, TX, USA, unmodified), proved that resveratrol can prevent and improve cardiac fibrosis and insufficiency caused by pressure overload by inhibiting the PTEN/AKT/Smad2/3 and NF-k B signaling pathway pathways, thus protecting the heart. In the isoproterenol-induced mice cardiac fibrosis model, Liu et al. [20] also found that resveratrol could reduce the myocardial fibrosis by inhibiting the activity of Smad2/3 (Figure 2). In vitro, Olson et al. [21], treating myocardial fibroblasts with resveratrol (5–25 microM, Fisher Scientific, Pittsburgh, PA, USA, unmodified), found that resveratrol can inhibit the proliferation and differentiation of cardiac fibroblasts. Resveratrol is an effective agonist of members of the sirtuin family, and the effects of its inhibition on the occurrence and development of cardiac fibrosis are closely related to the sirtuin family. Cappetta et al. [22], using rats as animal models, found that resveratrol could activate Sirtuin 1 (SIRT1) and interfere with myocardial fibrosis through TGF-β/Smad3 pathway. To improve the diastolic function of myocardium and reduce the myocardial remodeling caused by chemotherapy, Arafa et al. [23] also showed that SIRT1 can eliminate collagen synthesis and cardiac fibroblast differentiation induced by TGF-β stimulation. Moreover, resveratrol can influence the occurrence and progression of myocardial fibrosis by activating Sirtuin 3 (SIRT3), regulating the levels of hydroxyproline (Hyp), type I collagen gels (COL1), type Ⅲcollagen gels (COL3) and the expression of TGF- β, or by regulating the levels of SDF-1 and immune cytokines such as malondialdehyde (MDA), TNF-α, and IL-1 in oxidative stress [24]. In addition, resveratrol can also regulate other cytokines, such tissue inhibitor of metalloproteinases-2 (TIMP-2) and matrix metalloproteinase-2 (MMP-2). Matrix metalloproteinase (MMP) is an important enzyme system that regulates myocardial matrix metabolism, and it plays an important role in matrix degradation and collagen fiber synthesis [25]. It has been reported that resveratrol can inhibit the activity of MMP in human glioblastoma cells [9], but whether the protective effect of resveratrol on myocardial fibrosis is related to this principle remains to be further studied.

Resveratrol was originally used in cancer treatment and has displayed beneficial effects on most degenerative and cardiovascular diseases, including atherosclerosis [26,27,28], hypertension [29], ischemic heart disease [30], diabetes [31], aging [32], and myocardial hypertrophy [33]. Among them, the beneficial effect of resveratrol on cardiac hypertrophy is not only achieved by lowering blood pressure, but also involves other factors besides the change in hemodynamic load. These mechanisms may include the activation of anti-hypertrophic AMP-activated protein kinase (AMPK) signal pathway and the inhibition of hypertrophic Akt signal pathway [34]. AMPK (and its upstream kinase live kinase B1, LKB1) can not only antagonize hypertrophy, but also delay the transition from cardiac hypertrophy to heart failure. Importantly, AMPK can also inhibit cardiac remodeling by preventing myocardial fibrosis induced by angiotensin II [35].

The cardioprotective effect of resveratrol is related to a similar preconditioning effect with an enhanced adaptive response. Preconditioning is a protective and adaptive phenomenon. Transient ischemia and reperfusion (I/R) can make the heart resistant to subsequent ischemic injury [36]. Resveratrol preconditioning can produce cardioprotective effects when isolated hearts are subjected to 30 min of global ischemia followed by 2 h of reperfusion [37,38] or permanent occlusion of the left anterior descending coronary artery (LAD) [30]. During the pretreatment process, a low dose of resveratrol (0.5 to 1 mg/kg, Sigma–Aldrich, Saint Louis, MO, USA, unmodified; 10 microM, Sigma–Aldrich, Saint Louis, MO, USA, unmodified) produces an adaptive stress response, which induces the expression of cardiac protective genes and proteins (such as eNOS and cAMP response element binding protein, CREB) [39,40], leading to the phosphorylation and up regulation of Bcl-2 respectively [38], thus protecting cardiac tissue from cell death. Chen et al. [41] found that the anti-apoptotic effect of resveratrol may also be involved in the SIRT1-FOXO1 pathway. Interestingly, in the rat myocardial I/R model, resveratrol pretreatment can largely restore the altered microRNA in ischemic heart [42]. Resveratrol regulated miRNAs in I/R include miR-21, miR-20b, miR-27a and miR-9, which can regulate the extracellular-signal-regulated kinase (ERK), mitogen-activated protein (MAP) kinase signaling pathway in cardiac fibroblasts [43].

In general, the success of resveratrol treatment is based on its stimulating effect similar to that of any toxin: it exerts a beneficial effect at a low dose and a cytotoxic effect at a high dose [36]. Moreover, the so-called hormetic property of resveratrol may also be the cause of several controversial results associated with this molecule [44,45]. Although many studies have shown that resveratrol is a well-tolerated and safe compound in the human body [46,47], other studies have reported the dosing-time dependency [48] and the toxic effects of resveratrol in vitro and in vivo [44]. In particular, through the use of high-dose preparations, in vivo studies and clinical trials have shown that resveratrol and other drugs may have the potential for drug–drug interaction [49]. For example, resveratrol shows a systemic inhibition of cytochrome P450 (CYP) when taken at high doses [50,51]. In addition, resveratrol may lead to a decrease in first-pass metabolism, resulting in systemic exposure to certain CYP substrates given in combination [49]. Therefore, patients who ingest high doses of this food supplement in combination with other drugs may be at risk of clinically related drug–drug interactions, and these interactions may be harmful, because, in most cases, resveratrol may weaken the activity of these drugs [49]. The mechanism of myocardial fibrosis is complex and has not been studied clearly. Numerous studies have shown that resveratrol can delay the progress of myocardial and other tissue fibrosis to a certain extent, thus providing a potentially new strategy to inhibit cardiac fibrosis (Table 1).

### 3.1. Resveratrol Improves Adriamycin-Induced Cardiac Fibrosis

Adriamycin is a highly effective anthracycline chemotherapy drug and is the first-line drug for treating various cancers clinically. However, its cardiotoxicity limits its applications [62]. The mechanism of adriamycin-induced cardiotoxicity is complex, with cardiac fibrosis being an important event [23], involving several signaling pathways, including free radical generation, peroxynitrite formation, calcium imbalance, mitochondrial damage, apoptosis, and autophagy [63]. Adriamycin induces the production of oxygen free radicals in vivo, causing serious damage to the myocardial tissue. This long-term and chronic damage eventually lead to cardiac fibrosis. Similarly, adriamycin has been shown to induce cardiac fibrosis and inhibit cardiac function in a rat model. According to the literature, it not only upregulates the expression of TGF-β1 but also increases the levels of fibrosis markers in the left ventricular tissue and induces an excessive deposition of collagen fibers. Resveratrol can improve adriamycin-induced cardiotoxicity and cardiac fibrosis [23]. In addition, resveratrol, as an activator of histone deacetylase SIRT1, can activate SIRT1, interfere with the activation of cardiac fibroblasts through TGF-β/Smad3 pathway, reduce myocardial fiber formation, improve myocardial relaxation, and attenuate myocardial remodeling induced by doxorubicin [22].

Arafa et al. [23] have reported that resveratrol significantly reduced the left ventricular lipid peroxidation, hydroxyproline, and TNF-α levels, increased the activity of serum creatine kinase-MB (CK-MB) in adriamycin-treated rats, and prevented a decrease in heart weight in rats. As a pretreatment, a combination of resveratrol and adriamycin delayed the depletion of reduced glutathione and a decrease in superoxide dismutase activity in the left ventricle of rats. In addition, resveratrol improved adriamycin-induced upregulated expression of caspase-3 and TGF-β1 in the left ventricle, as well as the pathological changes such as necrosis and fibrosis in the left ventricle. Altogether, Arafa et al. [23] showed that resveratrol exerted a significant protective effect on adriamycin-induced cardiotoxicity and fibrosis in rats.

### 3.2. Resveratrol Improves Cardiac Fibrosis Induced by Atherosclerosis

During atherosclerosis, certain phenomena such as cardiac fibrosis, the increased expression of MMP-2, and the decreased expression of tissue-type MMP inhibitor-2 occur [25]. Resveratrol can alleviate an imbalance in the expression of MMP-2/tissue-type MMP inhibitor-2 and improve cardiac fibrosis. Under the action of various pathogenic factors, collagen synthesis occurs at a higher rate than degradation, which is one of the primary causes of cardiac fibrosis. Studies have shown that, during the formation of atherosclerosis (As), myocardial collagen undergoes accumulation, with type I collagen as the primary component, and the ratio of type I/III collagen increases, resulting in cardiac fibrosis [64]. MMP-2 is the key enzyme for the degradation of collagen into small polypeptides, which, under physiological conditions, is in relative equilibrium with its specific inhibitor called TIMP-2 [65].

During the occurrence and development of As, spasm, stenosis, and blockage of coronary artery and coronary arteriole can lead to myocardial cell injury and necrosis, finally causing repair fibrosis. A simultaneous overactivation of the reactive oxygen molecules and increased levels of inflammatory factors during the As process and increasing evidences show that oxidative and inflammatory factors play an important role in cardiac fibrosis caused by As. Ross [66] stated that As is an inflammatory disease, and that its inflammation involves the whole layer of blood vessels, including arterial adventitia. Hyperlipidemia leads to the dysfunction of coronary artery endothelium in the myocardium. Oxidized lipoproteins pass through the arterial wall and capillary wall adventitia and accumulate in the myocardial interstitium. This results in the increased recruitment of inflammatory cells and local production of cytokines and growth factors, especially TGF-β in the myocardial interstitium, stimulating phenotypic changes in cardiac fibroblasts and increasing collagen synthesis. Some scholars believe that cardiac fibrosis is a chronic inflammatory process [67]. Overactivated angiotensin and aldosterone in fibrosis play an important role in the inflammatory reaction [68]. Moreover, coronary endothelial dysfunction and imbalance in the secretion of endothelial-derived factors during As, such as decreased secretion of nitric oxide, prostacyclin, and bradykinin, and increased secretion of endothelin.

Studies have reported that the coronary artery endothelium can regulate the metabolism of cardiac myocytes and cardiac fibroblasts. For example, nitric oxide inhibits collagen synthesis in a concentration-dependent manner; prostaglandin can reduce collagen synthesis in cultured cardiac fibroblasts in a concentration-dependent manner and can significantly increase the activity of MMP-1. Bradykinin inhibits collagen proliferation and endothelin affects cardiac fibrosis [69,70]. In addition, MMPs can directly degrade collagen in the ECM, resulting in a disordered myocardial arrangement. They can increase chemotaxis, promote fibroblasts to enter the area after its action and cause collagen deposition [71]. MMPs mediate the activation of several substances, such as TNF-α, TGF-β, and IL-1, which, on the one hand, increase the synthesis of new collagen and destroy myocardial cells and, on the other hand, increase the expression of MMPs, constituting a vicious circle and further aggravating cardiac fibrosis. Resveratrol has antioxidant, anti-inflammatory, anti-As, and antifibrosis effects [72]. In addition, anti-angiogenesis and anti-tumor activities of resveratrol are closely related to its ability to significantly reduce the expression of MMPs and inhibit its gelatin-cracking activity [9,73]. Resveratrol can reduce the expression of MMP-2 and increase the expression of TIMP-2, thereby regulating the ratio of MMP-2/TIMP-2 and preventing cardiac fibrosis during As.

### 3.3. Resveratrol Improves Viral Myocarditis-Induced Cardiac Fibrosis

After treatment, although the majority of patients with myocardial collagen viral myocarditis (VMC) in the acute stage can be cured, the condition of a few patients turns chronic, and those with dilated cardiomyopathy (DCM) have a poor prognosis. Autopsy or endomyocardial biopsy data confirmed that the main pathological feature of VMC in the chronic phase was cardiac interstitial fibrosis, with a large amount of collagen type I and III deposited in cardiac interstitium. Thus, it affects the systolic and diastolic functions of the heart and becomes an important pathological mechanism of DCM formation [74,75]. Under physiological conditions, collagen plays an important role in maintaining the geometric configuration of the heart, ensuring the coordination between myocardial contraction and relaxation, and repairing myocardial injury. The primary factors determining myocardial compliance are tension and the content of collagen [76,77]. Moreover, under physiological conditions, collagen synthesis is primarily regulated by collagen gene mRNA transcription, whereas collagen degradation is regulated by several collagenases. Pathological hyperplasia of myocardial collagen can seriously affect myocardial compliance and decrease cardiac function. From a pathological point of view, collagen proliferation includes two processes, namely, restorative fibrosis and reactive fibrosis. Repairing fibrosis is a protective mechanism of the body, which is conducive to maintaining a normal myocardial function, whereas reactive fibrosis is the result of uncontrolled collagen expression. Pathological hyperplasia is an important step in the development of myocarditis into cardiomyopathy [78,79]. It can be described as a dynamic process, from an acute stage to a convalescent stage to a chronic stage of VMC, in which the repair fibrosis develops into a combination of repair fibrosis and reactive fibrosis, finally developing into reactive fibrosis [80].

Wang et al. [53] established a chronic VMC animal model in BALA/c mice infected with coxsackievirus B3 (CVB3). It was found that the myocardial collagen volume fraction in the resveratrol group was significantly lower than that in the control group; and, compared with the untreated VMC group, the serum concentrations of procollagen type I carboxyl-terminal peptide (PICP) and amino terminal peptide of type III procollagen (PIIINP) in the VMC group with high and medium doses of resveratrol were significantly decreased, and the level of amino terminal pro peptide of type I procollagen (PINP) increased significantly. When type I procollagen is secreted out of the cells by cardiac fibroblasts, the N-terminal and C-terminal lengthening peptides are cut by protease. In addition to a small amount deposited in the myocardial matrix, a large number of procollagen enters the blood circulation and eventually be metabolized by the liver. However, after treatment with resveratrol, the concentration of PINP detected in the blood increased significantly, indicating that type I procollagen was degraded, and it has also been suggested that resveratrol can inhibit myocardial collagen proliferation in VMC model mice and play an anti-myocardial fibrosis role. Li et al. [81] used resveratrol to treat CCD-18Co cells (CRL 1459), and found that resveratrol could inhibit collagen I synthesis in IGF-1-stimulated fibroblasts by inhibiting IGF-1R activation and activating SIRT1. Moreover, cardiac fibroblasts play a very important role in the formation of myocardial collagen [82]. Olson et al. [21] found that resveratrol (5–25 μM, Fisher Scientific, Pittsburgh, PA, USA, unmodified) can prevent cardiomyocytes from turning into myofibroblasts by inhibiting the proliferation and differentiation of cardiac fibroblasts (CFs). The mechanism is that resveratrol inhibits the activity of ERK 1/2 and ERK kinase induced by angiotensin II (ANG II), and inhibits ERK phosphorylation, thus inhibiting the proliferation of CFs cells.

### 3.4. Resveratrol Improves Alcohol-Induced Cardiac Fibrosis

The long-term heavy intake of alcohol can lead to serious damage to cardiac function and structure, resulting in increased proliferation and secretion of matrix collagen fibers by cardiac fibroblasts, forming myocardial fibrosis and promoting the process of alcoholic heart damage. This eventually leads to heart failure and various arrhythmias [83]. However, the underlying mechanism remains unclear. Alcohol can cause myocardial cell damage; however, it can promote the deposition of myocardial interstitial fibers as well. These two factors jointly promote myocardial structural damage, eventually causing end-stage heart failure and various serious arrhythmias [84]. Myocardial cell apoptosis or necrosis is followed by a repair process to regenerate the injured tissue [85]. However, alcohol-induced invasive injury reduces the chances of heart regeneration, resulting in ineffective repair mechanisms, which may lead to progressive fibrosis [86,87]. In fact, ethanol reduces the regeneration ability of myocardial cells and increases the fibrosis process [88]. Subendocardial and interstitial fibrosis gradually appear in the late stage of alcoholic cardiomyopathy (ACM) [89]. More than 30% of the ventricular fraction of myocardial cells can be replaced by fibrotic tissue, thus reducing the elasticity and contractility of the heart [90]. Certain myocardial cytokines, such as fibroblast growth factor 21 (FGF21), may regulate alcohol-induced cardiac fibrosis. For instance, FGF21-deficient mice showed higher blood pressure, more severe vascular inflammation and fibrosis, and changes in vascular function and vascular oxidative stress after angiotensin II perfusion [91]. Moreover, in HepG2 cells, resveratrol and SRT1720 increased the transcription activity of the FGF21 promoter and the level of FGF21 messenger RNA and protein, respectively [92].

MMP constitutes an important enzyme system that regulates myocardial matrix metabolism. On the one hand, the deposition of interstitial collagen leads to excessive collagen production; on the other hand, the degradation of collagen is inhibited. Therefore, MMPs not only play a role in the degradation of the matrix but also participate in the regulation of collagen synthesis. The final result is that MMP expression is often increased with increased fibrosis [93]. Therefore, some scholars have proposed to use MMP inhibitors to prevent myocardial remodeling. Cardiac fibrosis is accompanied by the destruction of the normal fibronectin (FN) skeleton structure [94]. Resveratrol can inhibit the expression of MMP in human glioblastoma cells [9]. Gelatinases include gelatinases A (MMP-2) and B (MMP-9), both of which can degrade interstitial proteins. MMP-2 and MMP-9 are involved both in the degradation and synthesis of matrix fibers. Previous studies have shown that with the deterioration of cardiac function, the expression and activity level of MMP increase [95]. However, the use of angiotensin receptors antagonist against myocardial remodeling may be accompanied by a decrease in MMP expression, indicating that the expression of MMP is positively correlated with myocardial remodeling. Moreover, alcohol can significantly increase the expression of MMP-2 [96], suggesting alcohol as one of the sub-mechanisms of alcoholic myocardial injury. Therefore, the inhibition of the expression and activity of MMP-2 and MMP-9 could be an effective preventive and treatment strategy for alcoholic myocardial damage. Therefore, inhibiting the overexpression of MMP-2 and MMP-9 could be the underlying molecular mechanism of resveratrol against alcoholic cardiac fibrosis. However, whether MMP and other inflammatory markers can be used as targets for the diagnosis and treatment of alcoholic cardiac fibrosis needs further studies.

### 3.5. Resveratrol Improves Diabetes-Induced Cardiac Fibrosis

Diabetes mellitus (DM) is a common metabolic disease, with cardiovascular disease being the primary cause of death of diabetic patients. Increasing evidence shows that dilated cardiomyopathy (DCM), which is characterized by early diastolic dysfunction and late systolic dysfunction, is independent of hypertension and coronary heart disease. Furthermore, it is one of the primary causes of heart failure in diabetic patients. Despite extensive research, a thorough understanding of its pathogenesis is still elusive, with no effective treatment available. Although the pathogenesis of DCM is complex and diverse, cardiac fibrosis is known to be involved. One of the characteristics of cardiac fibrosis is that, under the action of fibrogenic growth factors, especially TGF-β, matrix proteins start accumulating. For example, periostin, a matricellular protein, is known to regulate fibrosis formation in several diseases such as heart failure [97,98], myocardial infarction [99], and idiopathic pulmonary fibrosis [100]. Periostin can be stimulated by TGF-β and can regulate the expression of several downstream proteins, including α smooth muscle actin (α-SMA) and collagen, involved in fibrosis [52,101]. In addition, the activation of the ERK/TGF-β pathway and the upregulation of collagen production are involved in fibrosis [15]. Considering the relationship between TGF-β and periostin, the activation of the ERK/TGF-β/periostin pathway by oxidative stress is speculated to be one of the key events in the occurrence and development of cardiac fibrosis in dilated myocardial infarction. For example, Wu et al. [52] reported that periosteal protein is the core element of diabetes-related cardiac fibrosis, and that resveratrol can prevent its occurrence by inhibiting the ROS/ERK/TGF-β pathway.

Oxidative stress is an important sign of diabetes. For example, hyperglycemia increases ROS production by inducing glucose oxidation and producing mitochondrial superoxide. Qin et al. have shown that resveratrol (130 mg/kg/d, Orchid Chemicals and Pharmaceuticals, Nungambakkam, Chennai India, unmodified) can prevent DCM fibrosis by inhibiting oxidative stress in male C57BL/6J mice [102]. The increased feedback of ROS-inhibited aerobic oxidation of glucose promotes the anaerobic oxidation of glucose, i.e., enhanced glycolysis, thus increasing the production of diacylglycerol (DAG) and finally activating the DAG–PKA signaling pathway. The activation of the DAG–PKA signaling pathway plays an important role in the occurrence of cardiac fibrosis. In streptozotocin-induced diabetic pig myocardium, Guo et al. [103] found that myocardial protein kinase C (PKC) expression increased. It is suggested that PKC-β2 may be an important target of cardiovascular system injury in diabetes mellitus. Interestingly, Way’s research [104] found that PKC-β2 transgenic mice have myocardial fibrosis. Moreover, Giordo et al. [105] found that the inhibitory effect of resveratrol on PKC in human retinal endothelial cells induced by high glucose could counteract the NOX-mediated transformation from endothelial cells to mesenchymal cells. Therefore, resveratrol inhibits the overexpression of PKC-β2, which is considered as one of the important mechanisms to protect the morphology and function of myocardial cells and resist DCM myocardial fibrosis.

## 4. Conclusions

To summarize, although resveratrol exerts an antifibrosis effect via several growth factors, cytokines, and cell signaling pathways, and has several pharmacological effects, including antifibrosis, anti-inflammatory, antioxidative, lipid-lowering, and hypoglycemic effects, the research on its role in cardiac fibrosis is insufficient and warrants further exploration. Moreover, while resveratrol has great potential for clinical applications, several theoretical and experimental studies are still in the early stages involving animal research, and clinical drug research needs further progress. However, it is undeniable that resveratrol can delay the process of myocardial fibrosis to some extent, which may be a new method to inhibit myocardial fibrosis.

## Figures and Tables

**Figure 1 molecules-26-06860-f001:**
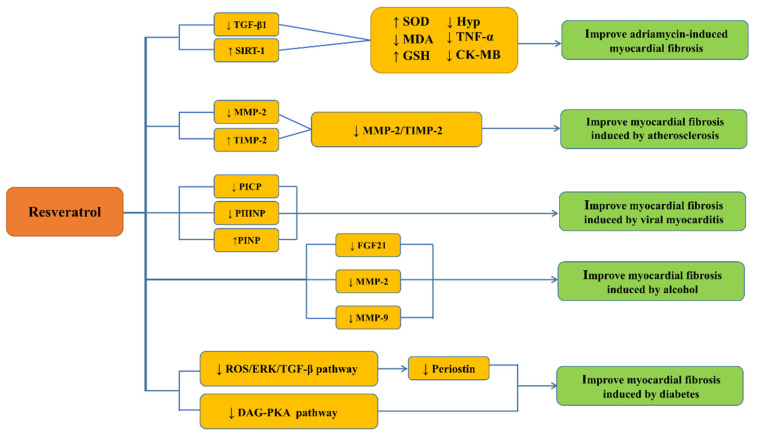
Potential mechanisms of resveratrol against cardiac fibrosis. ↓: a decrease; ↑: an increase. Transforming growth factor β1 (TGFβ1), Sirtuins-1(SIRT-1), superoxide dismutase (SOD), malondialdehyde (MDA), glutathione (GSH), phenolic oxidative coupling protein (Hyp), transforming growth factor α(TGF-α), creatine kinase-MB(CK-MB), matrix metalloproteinase-2 and 9 (MMP-2 and 9), tissue inhibitor of metalloproteinases-2 (TIMP-2), carboxyterminal propeptide of type I procollagen(PICP), amino-terminal propeptide of type III procollagen (PIIINP), aminoterminal propeptide of type I procollagen (PINP), fibroblast growth factor 21(FGF21), reactive oxygen species (ROS), extracellular signal-regulated kinase (ERK), diacylglycerol (DAG), protein kinase A (PKA).

**Figure 2 molecules-26-06860-f002:**
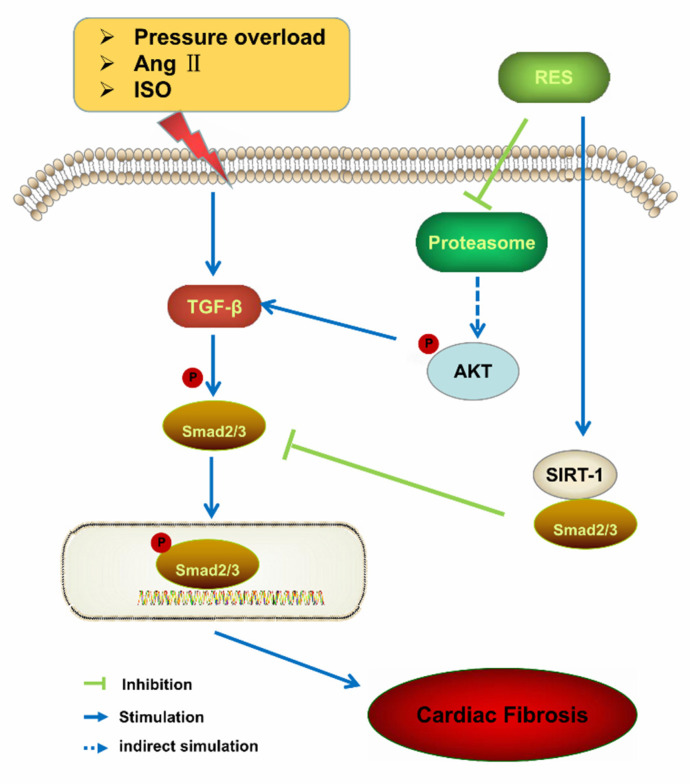
Proposed signaling pathway for the effect of resveratrol against myocardial fibrosis [19,20,22]: isoproterenol (ISO), resveratrol (RES), transforming growth factor β (TGFβ), Sirtuins-1(SIRT-1), angiotensin II (Ang II), protein kinase B (AKT).

**Table 1 molecules-26-06860-t001:** Effects of resveratrol on different types of fibrosis (↓: a decrease; ↑: an increase).

Animal	Cause of Fibrosis	Effective Dose	Effects or Mechanisms	Ref.
Wistar albino rats	Doxorubicin-induced cardiac fibrosis	20 mg/kg/d (4 weeks, p.o.)	Left ventricle:↓TNF-α, ↓TGF-β, ↓Hyp, ↓caspase-3	[23]
C57BL/6 mice	STZ -induced diabetic cardiac fibrosis	5 or 25 mg/kg/d (2 months, i.g.)	Suppression of ROS/ERK/TGF-β/periostin pathway	[52]
Balb/c mice	Chronic virus-induced cardiac fibrosis	10, 100 and 1000 mg/kg/d (30 days, i.g.)	Serum: ↓PICP, ↓PIIINP, ↑PINP	[53]
C57BL/6 mice	Isoproterenol-induced cardiac fibrosis	20 mg/kg/d (14 days i.p.)	Suppression of TGF-β/Smad2/3 pathway	[22]
ICR mice	LPS-induced pulmonary fibrosis.	0.3 mg/kg/d (4 weeks i.p.)	Suppression of TGF-β1/Smad pathway and oxidative stress	[54]
Sprague Dawley rats	BLM-induced pulmonary fibrosis	60 mg/kg/d (4 weeks i.p.)	Regulate miR-21 through MAPK/AP-1 pathway.	[55]
Wistar rats	NDMA-induced liver fibrosis	10 mg/kg/d (three consecutive days of each week for three weeks i.p.)	Suppression of oxidative stress and inhibit HSC activation (↓α-SMA, ↓MDA, ↑SOD, ↓carbonyls and ↑ATPases)	[56]
Wistar rats	CCl4-induced liver fibrosis	10 or 20 mg/kg/d (2 weeks, i.g.)	Reduce portal pressure and inhibit HSC activation (↓α-SMA, ↓collagen-1, ↓TGF-β,↓NF-κB)	[57]
C57BL/KS db/db mice	0.5% carboxymethyl cellulose sodium salt-induced kidney fibrosis	40 mg/kg/d (12 weeks, p.o.)	Suppression of AMPK/NOX4/ROS pathway	[58]
Sprague Dawley rats	UUO-induced kidney fibrosis	20 mg/kg/d (7 days, i.g.)	Suppression of the MAPK, PI3K/Akt, Wnt/β-catenin, and JAK2/STAT3 pathways	[59]
Male outpatients	Diagnosis of NIH type IIIa variant fibrotic	RSV 19.8 both one tablet twice daily (2 months, p.o.)	↑ Prostate volume secreted, ↓white blood cells counts	[60]
Human primary fibrotic BMFs	BMFs secrete	25, 50, and 100 µM (5 days, medium addition)	↑EZH2/H3K27me3, ↑miR-200c, ↓ZEB1	[61]

LPS: lipopolysaccharide; p.o.: per os; i.g.: intragastric; i.p.:intraperitoneal; BLM: bleomycin; STZ: streptozocin; VMC: viral myocarditis; NDMA: N’-nitrosodimethylamine; HSC:hepatic stellate cell; MDA: malondialdehyde; SOD: superoxide dismutase; AMPK: adenosine monophosphate activated protein kinase; UUO: unilateral ureteral obstruction. NIH: national Institutes of Health; BMFs: buccal mucosal fibroblasts; EZH2, zeste homolog 2; H3K27me3, trimethylated lysine 27 of histone H3.

## Data Availability

Not applicable.

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
