# Peer review of "Resveratrol against Cardiac Fibrosis: Research Progress in Experimental Animal Models"

_molecules, 2021, doi:10.3390/molecules26226860_

Round 1

Reviewer 1 Report

Title: Research progress of resveratrol against cardiac fibrosis

Journal: Molecules

Date: October 21st, 2021.

Contribution to the field:

The review addresses an important issue which is the presence of myocardial fibrosis induced by different factors: medications, viruses, heart attack, hypertension, diabetes, alcohol, and the anti-cardiac fibrosis effect of resveratrol.

The literature cited is robust and is an important source for consultation.

Major concerns:

The title should mention that the research progress is based in experimental animal models.

There was little attention to the writing of the manuscript:

-Sentences are repeated or meaningless; some sentences start with a lowercase letter.

-References are numbered incorrectly making it difficult to find the correct reference. After changing the numbers to the correct order I could not find the numbers cited in the table.

-There is no Manar et al as cited in the subtitle 3.1. The correct reference is Arafa et al.

-Table 1 should be omitted as it does not contain relevant information.

-Sometimes when the authors mention the word studies they cite only one reference:

 “Furthermore, studies have found that resveratrol can inhibit the activity of MMP in other tissues and cells [2], e.g., in the myocardium, thus resisting cardiac fibrosis”.

In some subtitles there is emphasis on mechanisms related to the formation of myocardial fibrosis but a less detailed description of resveratrol action targets, which is the object of the manuscript.

It should be shown that despite the different triggers that result in cardiac fibrosis there are several converging points in resveratrol's action.

Reviewer 2 Report

The review paper by Yu et al. is clearly written and easy to follow, however, it can be improved by providing more details. 

First of all, there is something wrong with the references in the paper. The references in the text do not match those in the reference list. The list shows 87 references and there are not that many in the paper text. This needs to be corrected. 

Most of the studies cited in this manuscript are obtained using mice or rats. This is interesting, however, studies that deal with humans, if there are more of them, could be cited and explained as well. Through the manuscript, the authors could emphasize if the results are in vitro or in vivo obtained, and on what animals or cell lines. Also, just like in Table 1, concentrations of resveratrol used in all cited studies should be mentioned. It should also be mentioned how resveratrol was applied and in what form, whether is unmodified, chemically modified, in combination with some other active molecules, as emulsion, nanoparticles... in general some more details for the method of application of resveratrol. 

There are some typos in the manuscript. Many sentences start with small letters. Please correct those. 

The last sentence in Conclusion: However, because resveratrol is present in several plants and can be easily extracted, and has high research prospects with potential anti-cancer, anti-oxidation, and anti-fibrosis effects, is not clear and should be rephrased. 

The last paragraph on page 3 is only text with no references. Please cite some references here about mentioned effects of resveratrol in this paragraph. 

Explanation of abbreviations in Figure 1should be placed in the Figure 1 legend.

Section 3.3 needs more citations, especially about the collagen. 

Reviewer 3 Report

In this manuscript, the authors review the research progress on the role of resveratrol against cardiac fibrosis.

Despite the compelling topic and the usefulness of summarizing the research progress in this field, in my opinion, this manuscript has several problems

First of all the manuscript should have page and line numbers in order for the reviewer to provide the related info in appropriate and specific way.

The manuscript looks as a draft rather than a final paper; there are many grammar mistakes and many sentences are incomplete or repeated.
Overall, the composition is confusing and hard to read since most of the sentences are not properly linked in a logical way to provide a smooth understandable reading. 
Different points are not clearly explained and are poorly supported by references which, in most of the cases, are wrongly reported and not really in line with the discussed topics.
Despite the aim of the work is reviewing the mechanisms of resveratrol against cardiac fibrosis (indicated in Fig 1), these last are poorly and insufficiently described in the text. 
In conclusion, this manuscript needs an extensive editing and reorganization. 

Round 2

Reviewer 1 Report

Minor concerns:

1- Line 88- “which promote fibroblasts to form and secrete ECM”

-fibroblasts secrete ECM????? Or, for example, secrete ECM macromolecules

2- Line 91- “can promote myocardial fibrosis and induce the expression of ECM” must be completed

-Suggestion: "induce the expression of ECM proteins".

3- It is not necessary to include the amount of resveratrol used and the manufacturer in each manuscript cited in the text.

4- Attention to the abbreviations: the noun together with the abbreviation must come in the first mention in the text.

Example: Line 121- …..TIMP-2 and MMP-2. Matrix metalloproteinase (MMP)

Also it is not necessary to  repeat the noun after the first mention in the text. 

5- Line 217. Rephrase the sentence:

“A simultaneous overactivation of the reactive oxygen system and increased levels of inflammatory factors during the As process and increasing evidence show that oxidative and inflammatory factors play an important role in cardiac fibrosis caused by several diseases.”

6- Line 284. Explain better:

“The level of amino terminal pro peptide of type I procollagen (PINP) increased significantly, suggesting that resveratrol can inhibit myocardial collagen proliferation in VMC model mice and play an anti-myocardial fibrosis role.-explain better

7- Repetition: Line 317: “And in HepG2 cells, In HepG2 cells, resveratrol a”

8- Conclusion:

8a- Rephrase the sentence: “However, resveratrol is present in several plants and can be easily extracted, which can delay the progression of fibrosis to some extent, which may be a new way to inhibit myocardial fibrosis.”

8b- “In addition, its clinical applications suffer from several shortcomings such as chronic toxicity, pharmacokinetics, and adverse reactions.” -Why it is mentioned in Conclusion, if is not cited in the main text.

9- It must be attributed to the references a sequential number according to the appearance in the text.

The last cited reference is number 88 but there are 97 references.

Reviewer 2 Report

The authors improved their manuscript significantly, and addressed all my issues. 

Reviewer 3 Report

The manuscript has been notably improved with the last editing and modifications. However, there are still some points that I believe the authors should incorporate in the manuscript.

Paragraph 2. Cardiofibrosis and its pathogenesis. Endothelial to mesenchymal transition (EndMT) is emerging as an important contributor in the development of fibrotic cardiovascular diseases as well as diabetes-associated cardiac fibrosis (PMID: 17660828, PMID: 32851865 ). Please include such an aspect in this paragraph, also because emerging studies showed the ability of resveratrol to counteract fibrosis by EndMT inhibition (e.g. PMID: 33540918)

Paragraph 3. Anti-cardiac fibrosis effect of resveratrol, line 160-162. The authors mentioned the cytotoxic effect related to high doses of resveratrol, but this aspect needs to be better clarified. Resveratrol indeed exhibits biphasic dose-dependent effects acting as an antioxidant at low concentrations and as a pro-oxidant at high concentrations. Moreover, resveratrol can also interact with the cellular redox state and with other drugs modifying its action and that of the other drug. I believe authors are aware that both negative and positive aspects of the discussed topic should be highlighted for the sake of information. Please include this aspect and see the following references for your information (PMID: 21264071, PMID: 32197410¸PMID: 19695122)

Paragraph 3.5 Resveratrol improves diabetes-induced cardiac fibrosis. I believe the authors should mention this recent paper as an example of the antifibrotic molecular mechanism of resveratrol during diabetes (PMID: 33540918)

Still, there are some issues with the references. The last reference in the text is number 88 but the references at the end of the report 97 articles.
